# Leadership development as a novel strategy to mitigate burnout among female physicians

Dawn M. Sears[ID][1☉*], Alexis Bejeck[2☉], Laurel Kilpatrick[2☉], Nicole Griggs[1‡], Lindsey Farmer[3‡], Brittany Jackson[4‡], Hania Janek[1‡], Anthony C. Waddimba[ID][5,6,7☉]

1 North Texas VA Medical Center, Dallas University of Texas Southwestern Medical Center, Dallas, Texas, United States of America, 2 Department of Internal Medicine, Scott and White Medical Center, Baylor Scott and White Health, Temple, Texas, United States of America, 3 Department of Internal Medicine, UTHealth McGovern Medical School, Houston, Texas, United States of America, 4 Department of Anesthesiology, Vanderbilt University, Nashville, Tennessee, United States of America, 5 Department of Surgery, Baylor University Medical Center, Baylor Scott and White Health, Dallas, Texas, United States of America, 6 Baylor Scott and White Research Institute, Baylor Scott and White Health, Dallas, Texas, United States of America, 7 Department of Medical Education, Texas A&M College of Medicine, Dallas, Texas, United States of America

☉ These authors contributed equally to this work.
‡ NG, LF, BJ, and HJ also contributed equally to this work.
* dawnsears@dawnsearsmd.com

## Abstract

### Background

Female physicians are more likely to experience burnout and less likely to hold leadership positions. Effective interventions are needed to support women physicians in the workforce.

### Objective

To determine if a shared learning, social-based leadership development program will impact burnout and career trajectory for female physicians.

### Design

Cohort study.

### Setting

Multispecialty healthcare system and state medical society members.

### Participants

Burnout and Engagement surveys were emailed to 5000 physicians within the Baylor Scott & White Health System (BSWH). The external control group consisted of 516 female physicians within the Texas Medical Association (TMA) and not associated with BSWH.

**Data availability statement:** All relevant data are within the manuscript and Supporting information files. We confirm that all raw data con be found in tables and supplements.

**Funding:** The Physicians Foundation grant number 3262215 awarded a multi-year grant to Baylor Scott & White Central Texas on 2/13/2018 to support the project titled Leadership Development as a Unique Tool to Fight Burnout in Women Physicians. DMS was awarded the grant for support of the study and received no salary support. The funders had no role in study design, data collection and analysis, decision to publish or preparation of the manuscript. www.physiciansfoundation.org.

**Competing interests:** No authors have competing interests.

Internal controls included both male (670) and female physicians (240) who did not participate in the program.

## Intervention

The Women Leaders in Medicine (WLiM) program included twice-annual in person summits and support programs throughout the 2-year study period.

## Measurements

The Maslach Burnout Index (MBI) was utilized to evaluate burnout. Surveys were conducted at three separate points and included interest in leadership, intent to retain current employment, and open comments.

## Results

Participants in WLiM had decreased frequency of high emotional exhaustion (mean 2.9 decreased to 2.5), decreased occurrence of high depersonalization (mean 1.6 decreased to 1.3), and improved levels of personal accomplishment (mean 4.7 improved to 5.1) and leadership aspiration (mean 7.4 to 7.8). Intention to stay went from 4.0 to 4.1.

## Conclusions

Burnout can be improved, and leadership aspirations fostered with a group leadership development in a cohort of female physicians.

## Introduction

Nearly half of all United States medical students, residents, and physicians experience burnout [1–4], a syndrome characterized by emotional exhaustion, depersonalization, and a low sense of personal accomplishment [2]. Burnout has been cited as a significant contributor to attrition, increased medical errors and risk to patient safety, and a negative impact on professionalism. In addition, burnout is associated with increased substance abuse, depression, anxiety, and suicidal ideation among physicians [5–12]. For female physicians, an increased amount of home domestic duties compared to male physicians, contribute to burnout and workforce attrition [13–15]. Loss of physicians from the healthcare system has a negative impact on healthcare access [3,16] and can increase the burden on healthcare systems due to the cost of recruitment and onboarding [17].

Female physicians experience job-related burnout and depression more frequently and profoundly than their male counterparts [18–19]. Burnout prevalence among female physicians exceeds that of male physicians by 60% [18]. Factors contributing to increased female physician burnout include greater patient expectations, family responsibilities, and a discrepancy between time spent with patients and revenue generated [13,18,20–23]. Data suggests that female physicians have the same number of complex medical patients as male physicians, with the addition of higher percentage of patients with time-intensive psychosocial problems. Due to these patient characteristics, outpatient appointments with female physicians are 21% longer than appointments with male physicians [18]. Female primary care physicians receive 26% more electronic medical record messages from patients and 24% more messages from staff than their male colleagues [21]. Female patients are more likely to express satisfaction with their female providers only when they perceive the clinicians' communication style as

caring. In a computer-generated virtual physician encounter, non-caring communication style did not negatively impact patient's satisfaction for male physicians [24]. Consequently, female physicians must overcome gender stereotypes and disproportionate patient expectations that drive excess burnout [18].

Half of all physicians entering the workforce are female [22], and female physicians achieve lower mortality, complications, and readmission rates than male physicians [6,22,25]. Female physicians are also more likely to provide preventive care, adhere to clinical guidelines, and provide psychosocial counseling [26]. If all practicing male physicians achieved these outcomes, an estimated 32,000 fewer patients would die each year [25]. Patients rate female physicians as more empathetic than their male counterparts [27], and female physicians are less likely to face medico-legal suits [28].

Despite these statistics, female physicians have greater difficulty identifying faculty mentors or sponsors, experience fewer opportunities for leadership development, hold fewer leadership positions, and are at higher risk for social isolation [29,30]. Evidence suggests that leadership development programs, peer mentoring, and professional coaching can promote female leaders' personal and professional growth in medicine [31–38]. Social support from peers improves satisfaction and skills necessary for success in academic endeavors, including research publication [33]. Professional coaching reduces burnout in male and female physicians by 17.1% after 5 months [35]. The leadership qualities of a physician also impact the well-being and satisfaction of the individual physicians under that leader [35]. In a Mayo Clinic study of 2800 physicians, each 1-point increase in a leader's rating score is associated with 3.3% less burnout among their team [35].

Few studies have investigated the real-world effectiveness of leadership coaching interventions in reducing burnout. To our knowledge, no study has field-tested a leadership development intervention targeted at female physicians and documented a direct impact on professional well-being and burnout. Therefore, we hypothesize that a leadership development program focused on building peer relationships and leadership skills of female physicians will enhance workplace effectiveness and decrease burnout in this population [39]. The present study examines this hypothesis by testing the effectiveness of a leadership training program (Women Leaders in Medicine [WLiM]) via evaluation using a standardized burnout questionnaire as well as career trajectory intent of the intervention cohort compared with control groups that did not participate in the program.

## Materials and methods

### Study design

This was a survey-based, prospective, non-randomized, pre-/post-intervention, non-equivalent comparison group study. The Baylor Scott & White Central Texas Institutional Review Board reviewed our anonymous survey as presenting minimal risk to respondents, waived written informed consent requirements, and approved the study (protocol # 018-101). Respondents read a cover letter describing the study, the privacy/confidentiality of responses, and freedom to accept/decline participation in the study. Informed consent was indicated by opting to respond to the survey.

### Intervention

Details of the WLiM program, developed internally, are described elsewhere.[39] Briefly, it included bi-annual leadership training summits for female practitioners plus quarterly regional dinner clubs, an intranet-based networking resource, an interactive social media platform, and opportunities for social gatherings. The intervention program, implemented

from fall 2017 to spring 2020 (before the COVID-19 pandemic) was supported by extramural funding (The Physicians Foundation grant funding, $150,000). Baylor Scott & White supported this program by permitting physicians to use CME time to attend the events.

## Participants

The intervention cohort were female physician survey respondents who attended WLiM program events, which were optional and available to all employed or affiliated physicians with BSWH. The first comparison cohort included female physician survey respondents who did not attend WLiM. A second comparison cohort included male physicians at our institution who responded to the survey but did not attend the events. A third comparison cohort consisted of female physician members of Texas Medical Association (TMA) who were not affiliated with our institution and did not attend the events.

## Data collection

A hyperlink to the multi-dimensional questionnaire was e-mailed to physicians and advanced practice providers (APPs) employed by (n ~ 2000) or affiliated with (n ~ 4000) our health system and to 516 TMA female physicians practicing outside our institution (see appendix flowsheet for full details). The survey was conducted in line with CHERRIES guidelines [40]. Survey data were managed through the secure Research Electronic Data Capture (REDCap™) platform [41]. Hyperlinks were unique to each respondent and automatically disabled after survey completion. Survey questionnaires were distributed at three separate time points over two years with a two-week window allowed at each distribution for returning responses. Recruitment of respondents occurred 03/01/2018-04/02/2018 and 01/08/2020-01/29/2020.

## Survey measures

Data collected included demographics and valid measures of study outcomes. Three dimensions of burnout (emotional exhaustion [EE], depersonalization [DP], personal accomplishment [PA]) were assessed by single-[42–43] and three-item [44] abbreviations of the subscales of the Maslach Burnout Inventory™-Human Services Scale for Medical Personnel (MBI-HSS-MP) (the full licensed Inventory was purchased with grant funding) [45]. High emotional exhaustion is associated with depression, anxiety, relational distress and suicide. Depersonalization is a measure of callousness and can have a negative impact on patient care. Personal accomplishment is associated with job satisfaction and retention. Congruence with the work environment was assessed by the Areas of Worklife Survey (AWS) [46]. Leadership aspiration was assessed using a single Likert-style item ("To what extent do you aspire to be a leader?") rated on a 0-10 (low to high aspiration) spectrum. Organizational retention was assessed using one item ("Do you plan on staying in your organization for the next 5 years") rated on a scale of 0 ('plan to leave soon') to 5 ('definitely expect to stay'). Primary outcomes were burnout, organizational retention, and leadership aspiration at post-intervention follow-up.

## Statistical analysis

Data from physician respondents to the baseline (first) and post-intervention (third) surveys were analyzed. Descriptive analyses quantified continuous variables as means (± standard deviations [SD] or 95% confidence intervals [CI]) or medians (interquartile range [IQR]), based on normal vs skewed distribution in the sample, and discrete variables as percentages (%). We evaluated unadjusted group differences (a) in normally distributed continuous variables via the Student's t-test; (b) in continuous variables with a skewed distribution via the Mann-Whitney or Wilcoxon rank sum tests; and (c) in discrete variables via Pearson's

Chi-square test. Bivariate associations between ordinal scales were tested via polychoric correlations. Friedman's repeated measures analyses of variances (ANOVAs) by ranks test [47] evaluated group differences in how burnout, retention, or leadership aspiration changed from the baseline to the post-intervention follow-up. If Friedman's test rejected the null hypothesis of no differences, Dunnett's posthoc test [48] was done to confirm or disprove significant differences between any pair of groups. Nonparametric repeated measures multivariate analyses of variance (MANOVAs) followed similar steps. Multivariable logistic regressions were used to test multivariate associations of high burnout (per published cut-offs on single- [49] and 3-item [44] abbreviations of MBI-HSS-MP subscales) with female versus male physician group status. Multivariable models adjusted for physician demographics, practice attributes, and work-life domains. The significance threshold for statistical test was $p < 0.05$. Analyses were performed using SAS software version 9.4 (SAS, Inc.; Cary, NC).

## Results

The baseline survey in 2018 included 924 physicians and 322 APPs (Fig 1). The final survey in 2020 (prior to the COVID-19 pandemic in the United States) elicited 1,239 responses from physicians and 425 APPs (Fig 2). Data from APPs is not reported and analyzed in this manuscript. The capture rate was robust with 35.9% completing the entire first survey. Overall, survey response was evenly distributed by binary gender, with 670 male and 436 female physicians completing the entire 2020 survey (proportionate to gender of employment).

Participation in WLiM summits was high, with 90% of events reaching capacity (100 physicians in central Texas and 40 physicians in north Texas) and multiple events had waitlists. The initial goal was to enroll at least 25% of female physicians practicing at each BSWH location in the WLiM program. In central Texas, 41.3% (239 of 579) of female physicians attended events and 30.3% (137 of 452) female physicians in north Texas attended at least one WLiM event. Most participants attended three events.

Baseline demographics of BSWH female physicians who attended WLiM events versus non-participants was similar in age (both mean 41), marital status, ethnicity, child rearing

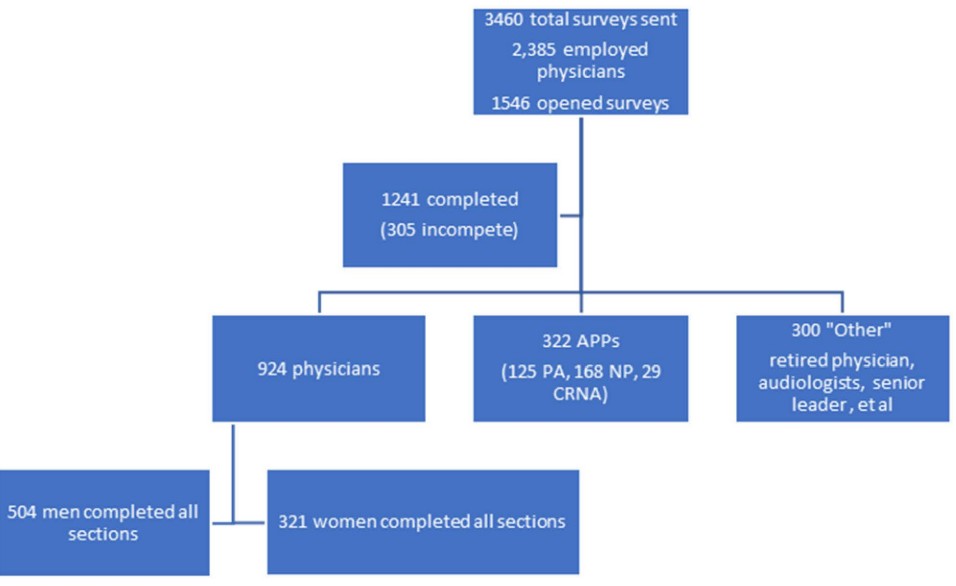

**Fig 1. Flowchart of responses- Baseline Survey from 2018- to all employed and affiliated physicians and APPs.**

status and FTE (full time equivalent) (74.8% vs 74.5% at 1.0 FTE). Early career physicians were more prevalent in non-attendees versus attendee cohorts (less than 5 years in practice 29% vs. 16.5%). Comparing BSWH female physician who attended WLiM events to female physicians in the TMA group, (control group, cohort 3), the TMA physicians were older than the WLiM group, less likely to be a parent of young or school-aged children, and more likely to have practiced more than 15 years (Table 1). The TMA and WLiM groups had similar work hours, with 74.8% of WiLM and 76.7% of TMA physicians working 40 hours per week. No significant difference existed between the groups' marital status or ethnicity.

At baseline, the number of respondents scoring "high" on the single-item Maslach Burnout Inventory Emotional Exhaustion (MBI-EE) item were: 12 of 31 (38.71%) of WLiM participants, 126 of 296 (41.89%) of female non-participant controls (Cohort 1), 163 of 544 (29.96%) of male physician controls (Cohort 2), and 191 of 434 (44.01%) of TMA female physician controls (Cohort 3). Of the comparison groups at our institution, burnout scores on the MBI-EE were significantly higher on the follow up survey for female physicians not participating in WiLM compared to WLiM participants (OR = 2.5r, 95% CI = 1.10-5.88; p = 0.022). After completing the program, WLiM participants had decreased incidence of emotional exhaustion (MBI-EE), decreased incidence of depersonalization (MBI-DP), and improved levels of personal achievement (MBI-PA) compared with non-participants (cohort 1) (Table 2). A longitudinal comparison of the 143 women physicians who completed all questions on surveys in 2018 and 2020 demonstrated that participants in WLiM events had lower MBI-EE scores (2.9 at baseline to 2.5 at follow-up) while non-participants showed a less significant decrease from 3.2 at baseline to 3.0 at follow-up. Improvement in single-item MBI-EE scores among the WLiM group was double the improvement seen in female non-participant controls (cohort 1).

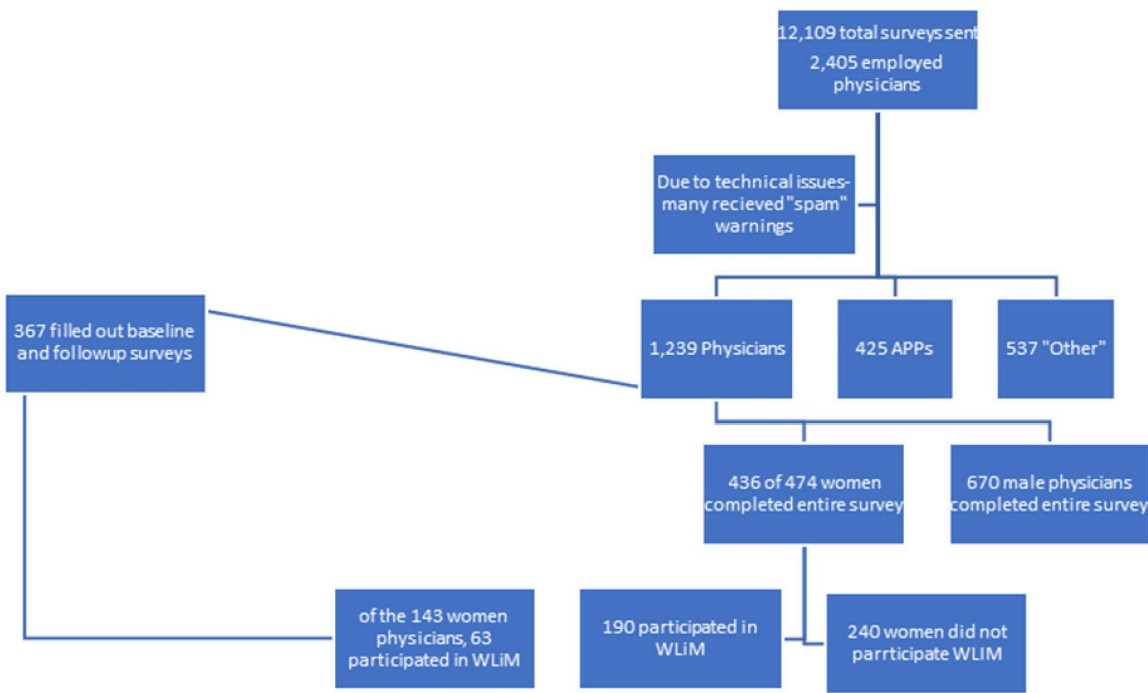

**Fig 2. Flowchart of responses- Follow up survey from 2020 connected to Employee Engagement system wide survey and went to all "Senior leaders" instead of only physicians and clinicians.**

**Table 1.** Participants demographics at baseline.

| Variable | Intervention Group | Cohort 1 | Cohort 2 | Cohort 3 |
|---|---|---|---|---|
| | BSWH Female Physicians Who Attended WLiM Events | BSWH Female Physicians Who Did Not Attend WLiM Events | BSWH Male Physician Respondents | Texas Medical Association (TMA) Female Physician Respondents |
| Overall sample, N | 190 | 240 | 670 | 516 |
| Age in Years, median (IQR) | 41 (37, 48) | 41 (35, 47) | 49 (41, 59) | 47 (40, 56) |
| Marital Status, n (%) | | | | |
| Married/ partnered | 122 (84.7%) | 216 (82.9%) | 550 (92.6%) | 262 (85.3%) |
| Single/never married | 14 (9.7%) | 23 (10.7%) | 28 (4.7%) | 43 (14.0%) |
| Widowed | 0 (0%) | 2 (0.9%) | 1 (0.2%) | 0 (0%) |
| Divorced or separated | 8 (5.6%) | 12 (5.6%) | 15 (2.5%) | 2 (0.7%) |
| Ethnicity, n (%) | | | | |
| White/ Non-Hispanic | 84 (60.9%) | 118 (59.3%) | 433 (76.4%) | 289 (71.4%) |
| Hispanic or Latino | 8 (5.8%) | 18 (9.0%) | 27 (4.8%) | 0 (0%) |
| Black/ African American | 11 (8.0%) | 9 (4.5%) | 6 (1.1%) | 40 (9.9%) |
| Asian | 27 (19.6%) | 45 (22.6%) | 75 (13.2%) | 58 (14.3%) |
| Middle Eastern | 1 (0.7%) | 2 (1.0%) | 10 (1.8%) | 0 (0%) |
| American Indian/ Alaskan | 0 (0%) | 0 (0%) | 0 (0%) | 0 (0%) |
| Hawaiian/ Pacific Islander | 0 (0%) | 0 (0%) | 0 (0%) | 2 (0.5%) |
| Multiple/ Mixed | 3 (2.2%) | 3 (1.5%) | 4 (0.7%) | 0 (0%) |
| Other | 4 (2.9%) | 4 (2.0%) | 12 (2.1%) | 16 (4.0%) |
| Child Rearing Status, n (%) | | | | |
| Child/ren <5 years old | 44 (26.5%) | 66 (28.3%) | 105 (14.5%) | 44 (15.9%) |
| School age child/ren | 77 (46.4%) | 102 (43.8%) | 286 (39.4%) | 103 (37.2) |
| College age child/ren | 25 (15.1%) | 26 (11.2%) | 118 (16.3%) | 23 (8.3%) |
| Adult child/ ren | 16 (9.6%) | 30 (12.9%) | 198 (27.3%) | 62 (22.4%) |
| Special Needs child/ ren | 4 (2.4%) | 9 (3.9%) | 18 (2.5%) | 1 (0.4%) |
| Child/ren passed away | 0 (0%) | 0 (0%) | 0 (0%) | 44 (15.9%) |

*(Continued)*

**Table 1.** (Continued)

| Variable | Intervention Group | Cohort 1 | Cohort 2 | Cohort 3 |
|---|---|---|---|---|
| | BSWH Female Physicians Who Attended WLiM Events | BSWH Female Physicians Who Did Not Attend WLiM Events | BSWH Male Physician Respondents | Texas Medical Association (TMA) Female Physician Respondents |
| Clinical Experience, n (%) | | | | |
| <5 years | 36 (16.5%) | 65 (29.0%) | 71 (12.0%) | 82 (15.9%) |
| 5–10 years | 45 (20.6%) | 62 (28.3%) | 128 (21.5%) | 127 (24.6%) |
| 11–15 years | 27 (12.4%) | 32 (14.6%) | 89 (15.0%) | 89 (17.3%) |
| 16–20 years | 14 (6.4%) | 25 (11.4%) | 77 (13.0%) | 83 (16.1%) |
| 21–30 years | 26 (11.9%) | 30 (16.5%) | 124 (20.9%) | 106 (20.5%) |
| >30 years | 2 (0.9%) | 5 (0.2%) | 104 (17.5%) | 38 (7.4%) |
| Full time FTE, n (%) | | | | |
| <0.5 FTE | 2 (1.4%) | 6 (2.7%) | 13 (2.2%) | 29 (6.3%) |
| 0.5–0.75 FTE | 16 (10.9%) | 29 (13.2%) | 19 (3.2%) | 33 (7.1%) |
| 0.75–0.95 FTE | 19 (12.9%) | 21 (9.5%) | 41 (6.9%) | 46 (9.9%) |
| 1.0 FTE | 110 (74.8%) | 164 (74.5%) | 521 (87.7%) | 355 (76.7%) |

Women physicians who participated in the events had similar demographics to those who did not participate. The comparison TMA group was older and had more child rearing similarities to the male physicians at BSWH.

Depersonalization improved in WLiM participants, with the single-item MBI-DP measure decreasing from a mean of 1.60 (95% CI 1.32-1.88) at baseline to 1.26 (0.83-1.70) at follow-up among WLiM participants (Table 2). In contrast, depersonalization scores worsened at follow up among male physicians, increasing from 1.59 (1.44-1.74) at baseline to 1.68 (1.49-1.88) at follow-up. Worsening depersonalization scores were also seen in the TMA control group (Table 2). For the item of personal accomplishment (MBI-PA), WLiM participants had an improvement in ratings on the single-item MBI-PA measure from 4.69 (4.44-4.95) at baseline to 5.06 (4.69-5.42) at follow-up. There was a numerical improvement in MBI-PA measures for WLiM participants that was seven times greater than observed with female physician non-participant controls (cohort 1) and twice that among male physician controls (cohort 2).

Participants in WLiM demonstrated an increase in leadership aspirations (survey item "To what degree do you aspire to be a leader") from 7.45 (7.04-7.85) at baseline to 7.81 (6.94-8.68) at follow-up among WLiM participants. For comparison, female non-participant controls (cohort 1) increased from 5.32 (4.89-5.76) to 6.23 (5.81-6.65) and from 6.64 (6.35-6.92) to 6.75 (6.34-7.16) among TMA controls (cohort 3) (Table 2). In contrast leadership aspiration ratings fell from 6.57 (6.32-6.83) at baseline to 6.42 (6.07-6.77) at follow-up among male physician controls (cohort 2) (Table 2). Female non-participant controls (cohort 1) experienced the most improvement.

Multivariable analysis was performed after adjusting for covariates. For participants in WLiM, there were statistically significant improvements in emotional exhaustion and depersonalization. In contrast, leadership aspiration decreased for women not participating in the program, while EE and DP worsened. Male physicians completing the survey had increasing EE and DP scores between the baseline and final surveys, but also reported an increased sense of PA. The women in the control group did have improved sense of PA. (S1 and S2 Tables). Nonparametric repeated measures ANOVAs (S1 Table) and MANOVAs (S2 Table) comparing

Table 2. Burnout, Leadership aspiration and organizational retention at follow up vs. baseline.

| Study Outcome | Intervention Group: BSWH Female Physicians Who Attended WLiM Events | | Cohort 1 BSWH Female Physicians Who Did Not Attend WLiM Events | | Cohort 2 BSWH Male Physician Respondents | | Cohort 3 Texas Medical Association (TMA) Physician Respondents | |
|---|---|---|---|---|---|---|---|---|
| | Mean (±SD) | Median (IQR) | Mean (±SD) | Median (IQR) | Mean (±SD) | Median (IQR) | Mean (±SD) | Median (IQR) |
| **Emotional exhaustion (EE)** | | | | | | | | |
| • "I feel burned out from work" at baseline | 2.9 (1.8) | 3 (1, 4) | 3.2 (1.8) | 3 (1, 5) | 2.6 (1.9) | 2 (1, 4) | 3.3 (1.9) | 3 (2, 5) |
| • "I feel burned out from work" at follow-up | 2.5 (1.8) | 2 (1, 4) | 3.0 (1.9) | 3 (1, 5) | 2.7 (1.9) | 2 (1, 4) | 3.2 (1.8) | 3 (2, 5) |
| **Depersonalization (DP)** | | | | | | | | |
| • "I've become more callous" at baseline | 1.6 (1.7) | 1 (0, 3) | 1.7 (1.8) | 1 (0, 3) | 1.6 (1.8) | 1 (0, 3) | 2.0 (1.8) | 1 (1, 3) |
| • "I've become more callous" at follow-up | 1.3 (1.2) | 1 (0, 2) | 1.5 (1.7) | 1 (0, 2) | 1.7 (1.8) | 1 (0, 3) | 2.1 (1.8) | 1 (2, 3) |
| **Personal achievement (PA)** | | | | | | | | |
| • "accomplished many worthwhile things" at baseline | 4.7 (1.5) | 5 (4, 6) | 4.5 (1.4) | 5 (4, 6) | 4.7 (1.4) | 5 (4, 6) | --- | --- |
| • "accomplished many worthwhile things" at follow-up | 5.1 (1.0) | 5 (5, 6) | 4.5 (1.5) | 5 (4, 6) | 4.8 (1.4) | 5 (4, 6) | --- | --- |
| **Leadership aspirations** | | | | | | | | |
| • "To what degree do you aspire to be a leader" at baseline | 7.4 (2.4) | 8 (6, 10) | 5.3 (3.2) | 5 (2, 8) | 6. 6 (3.1) | 7 (5, 10) | 6.6 (3.0) | 7 (5, 10) |
| • "To what degree do you aspire to be a leader" at follow-up | 7.8 (2.6) | 8 (6.5, 10) | 6.2 (3.1) | 7 (4, 9) | 6.4 (3.4) | 7 (3, 10) | 6.8 (3.2) | 8 (5, 10) |
| **Retention with Organization** | | | | | | | | |
| • "plan on staying in organization for 5 years?" at baseline | 4.0 (1.3) | 5 (3, 5) | 3.7 (1.5) | 4 (3, 5) | 4.0 (1.4) | 5 (3, 5) | 3.6 (1.7) | 4 (3, 5) |
| • "plan on staying in organization for 5 years?" at follow-up | 4.1 (1.9) | 4.5 (3.5, 5) | 4.1 (1.4) | 5 (3.5, 5) | 3.8 (1.6) | 5 (3, 5) | 3.6 (1.6) | 4 (3, 5) |

The parameters of burnout and leadership aspirations improved for the women who participated in WLiM events compared to those who did not. Both control groups scored worse on some of the burnout parameters. Intention to stay remained unchanged over time for TMA women physicians and worsened from male physicians. Intention to stay improved in women who attended and did not attend WLiM events at the institution.

the intervention group versus female and male controls confirmed this difference as statistically significant.

## Discussion

In this large, longitudinal study of physicians in Texas, we found that participation in the multi-component, WLiM program for female physicians was associated with improvement in emotional exhaustion and depersonalization dimensions of burnout as well as greater leadership aspirations. Although a significant amount of research has described the etiology of burnout among physicians, few studies have investigated interventions that proved beneficial across a large organization or population of practicing physicians. This study adds to the literature by describing the impact on burnout and leadership aspirations for female physicians after attending a series of professional development events that created opportunities for connection, networking and education.

Burnout is associated with being a female healthcare provider, working in primary care, having substantial domestic caregiving responsibilities, and carrying large student debt burden [3,13,18,50,51]. Factors that protect against burnout include participating in leadership development courses, serving in leadership roles, having a mentor and serving as a mentor [13,18,20,21,23]. With this knowledge, our goal was to create a program that fosters collegiality, belonging, and collective learning. By demonstrating a reduction in burnout symptoms and increase in leadership aspiration, we have introduced WLiM as a

reproducible program that positively impacts female physicians working in a large health-care organization.

At baseline, more than 1/3 of all physicians surveyed scored "high" on the single-item burnout inventory (MBI-EE), which is consistent with national data on physician burnout. The female physicians participating in WLiM events were the only group to demonstrate a statistically significant decrease in emotional exhaustion and, at follow up, WLiM participants demonstrated the lowest MBI-EE scores of all groups surveyed. BSWH female physicians who did not participate in WLiM (cohort 1) and TMA female physicians (cohort 3) maintained the highest MBI-EE scores, indicating ongoing risk of burnout in these populations. Furthermore, WLiM participants were the only group to show a statistically significant reduction in depersonalization scores (MBI-DP). Like the MBI-EE findings, MBI-DP was highest among the TMA female physician group (cohort 3) at baseline and follow up indicating they are at ongoing risk of burnout.

WLiM participants showed higher leadership aspirations on baseline surveys compared to the other groups. BSWH non-participant female physicians (cohort 1) scored lowest on this domain, which may indicate a bias toward WLiM participation for female physicians who are interested in leadership. Interestingly, early career physicians are a vulnerable group to burn-out and, unfortunately, were overrepresented in non-participants. In the comments section of our surveys, respondents noted that being sole bread winner and having young children may have been barriers for participation and this warrants further study and development to serve this population.

Recent consensus papers recommend that physician wellness interventions focus on a multimodal approach incorporating strategies such as personal wellness/self-care workshops and organizational/systemic level changes to electronic medical records, work hours, and workloads [52–56]. Because these factors have been indicated as contributors to burnout, their improvement could ameliorate personal burnout symptoms and decrease physician turnover at the organizational level [57,58]. Participants in courses addressing antecedents of burn-out have previously been shown to improve MBI (EE, DP, PA) scores, depression, anxiety, and stress [59–60]. Structured professional coaching programs have also shown a significant reduction in EE, imposter syndrome, and self-compassion [37,61]. While our program did not include validated surveys to measure self-compassion and imposter syndrome, we demonstrated an improvement in two domains of burnout and promoted leadership aspirations using the WLiM program.

Participant comment review (reported elsewhere) [39] demonstrated that the effectiveness of the WLiM program results from the social and personal connections to allow peer support, coping and stress management skills, personal wellness/self-care, and professional development to equip women physicians with the skills to advance to leadership positions. Burnout of leaders contributes to burnout of physicians under their leadership [35]. The positive impact of the WLiM program on women physicians could potentially generate a system-wide effect across an organization as women pursue leadership positions and remain engaged in their careers. Organizations are now becoming aware of the cost of attrition of women physicians. Increased representation of women leaders' voices could ultimately improve retention of women physicians as issues such as maternity leave, lactation accommodations, and flexible scheduling are addressed leadership and institutional support are required to create a WLiM program. Key stakeholders such as department chairs, Chief Medical Officers, Chief Wellness Officers, business leaders, administration, and nursing leaders were all involved in our program and contributed to its success.

This study has several limitations. First, physicians self-selected to participate in WLiM, which may limit generalizability of our results. Self-selection could also result in over

representation of women physicians who may have more interest addressing burnout concerns. WLiM participants showed higher leadership aspirations on baseline surveys compared to the other groups. BSWH non-participant female physicians (cohort 1) scored lowest on this domain, which may indicate a bias toward WLiM participation for female physicians who are interested in leadership. Randomization was not employed in this study. Despite the study occurring in large geographic area of over 300 square miles and including both rural clinics and large metropolitan hospitals, the study group was all employed by a single large medical organization, which could limit generalizability outside of that institutional context. In addition, this program did not occur in isolation. At a national level, the issue of workplace disparity for female physicians gained traction, and the social media themes of #TimesUpHealthcare and #MeToo were occurring simultaneously. Medical societies began emphasizing equality for women in medicine. Physicians in our study may have accessed other resources, such as personal coaches, therapists, and spiritual support, which can influence well-being [37,52,62]. Inter-group differences in attrition might contribute to selection biases. All participants had to actively open and fill out the survey. Finally, all survey data were collected before the coronavirus pandemic and therefore, our study reflects experiences prior to this international event. In summary, plausible threats to validity were history, maturation, attrition, selection, and/or regression to the mean [63].

A unique strength of this paper is the inclusion of both internal and external comparison groups, comparing three cohorts of female physicians and one cohort of men. Another unique strength is the collection of the full MBI for over a thousand physicians, with 2 years of follow up results. This study evaluated a practical real-world program that can help physician burnout and interest in leadership roles for female physicians.

## Supporting information

**S1 Table. Nonparmetric multivaiate repeated measures analysis of variance by ranks on the outcome scores.**
(PDF)

**S2 Table. Friedman's unadjusted repeated measures nonparametric analyses of variance by rank on the outcome scores.**
(PDF)

## Acknowledgments

We would like to thank all the physicians and APPs who took time to complete these surveys and BSWH human resources plus the Texas Medical Association for assistance with disseminating these surveys. This work was supported by resources provided by the Dallas VA Medical Center. VA/US Government Disclaimer: The contents do not represent the views of the U.S. Department of Veterans Affairs or the United States Government.

## Author contributions

**Conceptualization:** Dawn M. Sears, Lindsey Farmer, Hania Janek.

**Data curation:** Dawn M. Sears, Laurel Kilpatrick, Lindsey Farmer, Brittany Jackson.

**Formal analysis:** Nicole Griggs, Anthony C. Waddimba.

**Funding acquisition:** Dawn M. Sears.

**Investigation:** Dawn M. Sears, Laurel Kilpatrick.

**Methodology:** Dawn M. Sears, Laurel Kilpatrick, Anthony C. Waddimba.

**Project administration:** Dawn M. Sears.

**Resources:** Anthony C. Waddimba.

**Software:** Anthony C. Waddimba.

**Supervision:** Dawn M. Sears, Laurel Kilpatrick, Anthony C. Waddimba.

**Validation:** Dawn M. Sears, Alexis Bejeck, Laurel Kilpatrick, Anthony C. Waddimba.

**Visualization:** Dawn M. Sears, Alexis Bejeck, Laurel Kilpatrick.

**Writing – original draft:** Dawn M. Sears, Alexis Bejeck, Laurel Kilpatrick, Lindsey Farmer, Brittany Jackson.

**Writing – review & editing:** Dawn M. Sears, Alexis Bejeck, Laurel Kilpatrick, Anthony C. Waddimba.

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
