## [Decision Letter · Decision Letter 0]

2 Jan 2025

PONE-D-24-35973Leadership Development as a Novel Strategy to Mitigate Burnout Among Female PhysiciansPLOS ONE

Dear Dr. Sears,

Thank you for submitting your manuscript to PLOS ONE. After careful consideration, we feel that it has merit but does not fully meet PLOS ONE’s publication criteria as it currently stands. Therefore, we invite you to submit a revised version of the manuscript that addresses the points raised during the review process.

We look forward to receiving your revised manuscript.

Kind regards,

Francesco Marcatto, Ph.D.

Academic Editor

PLOS ONE

Journal Requirements:

2.  Please include a separate caption for each figure in your manuscript.

3. Please include your tables as part of your main manuscript and remove the individual files. Please note that supplementary tables (should remain/ be uploaded) as separate "supporting information" files"

Reviewers' comments:

Reviewer's Responses to Questions

**Comments to the Author**

1. Is the manuscript technically sound, and do the data support the conclusions?

Reviewer #1: Yes

Reviewer #2: Yes

2. Has the statistical analysis been performed appropriately and rigorously? 

Reviewer #1: Yes

Reviewer #2: Yes

3. Have the authors made all data underlying the findings in their manuscript fully available?

Reviewer #1: Yes

Reviewer #2: Yes

4. Is the manuscript presented in an intelligible fashion and written in standard English?

Reviewer #1: Yes

Reviewer #2: Yes

5. Review Comments to the Author

Reviewer #1: The manuscript titled "Leadership Development as a Novel Strategy to Mitigate Burnout Among Female Physicians" examines a critical issue in healthcare, focusing on the prevalence of burnout among female physicians and proposing the Women Leaders in Medicine (WLiM) program as a novel intervention. The study is well-structured and methodologically sound, providing a detailed evaluation of how leadership development can mitigate burnout and enhance career trajectories in this population. The intervention itself is comprehensive, incorporating bi-annual leadership summits, networking opportunities, and continuous support programs, which are designed to address burnout from multiple dimensions.

The study demonstrates considerable methodological rigor. It employs validated tools, such as the Maslach Burnout Index (MBI), to assess burnout dimensions and uses a combination of internal and external control groups to strengthen the robustness of the findings. The longitudinal nature of the data collection, spanning over two years with surveys conducted at three time points, adds depth to the analysis and ensures the reliability of the results. The findings reveal statistically significant improvements in emotional exhaustion, depersonalization, and personal accomplishment among participants of the WLiM program, as well as an increase in leadership aspirations compared to non-participants. These results underscore the potential of leadership development programs to not only reduce burnout but also empower female physicians in their professional growth.

Despite its strengths, the study has some limitations that could impact its generalizability. The self-selection of participants into the WLiM program introduces a potential bias, as those already motivated to seek leadership roles or address burnout may be overrepresented. Additionally, the study is confined to a specific healthcare system, which may limit its applicability to other institutional or cultural contexts. The reliance on self-reported data further raises concerns about response bias, and the non-randomized design restricts causal inferences. While the manuscript acknowledges these limitations, a more detailed discussion of their implications would enhance the study's transparency and credibility.

External factors, such as concurrent social movements like #MeToo and #TimesUpHealthcare, might have also influenced the outcomes, and the manuscript could benefit from further exploration of these potential confounders. Expanding the analysis to include demographic subgroups could provide additional insights into the intervention's differential impacts. Furthermore, a follow-up assessment of long-term outcomes would be valuable to determine the sustainability of the observed benefits.

Overall, this manuscript makes a significant contribution to the literature on physician burnout and leadership development. The intervention is practical, evidence-based, and addresses a pressing issue in healthcare. With minor revisions to address its limitations and to broaden the scope of its discussion, the manuscript is well-suited for publication and will provide valuable insights for healthcare institutions aiming to support their workforce and promote leadership among female physicians.

Have you bought the licence for Maslach Burnout Inventory?

Reviewer #2: I read this study with great interest given its relevance to many themes around important issues of burnout, retention, leadership, and gender disparities within these issues. Overall it is written clearly, contributes to the literature, and contains interesting information. The following comments and suggestions are to aid the authors in making sure the content is clear to readers.

Minor typos:

Abstract / Results paragraph:

Change “improved levels personal accomplishment…” to “improved levels (of) personal accomplishment…”

Results section / Paragraph beginning “Baseline demographics…”

Change “The TMA and WLiM groups had similar work hours, with 74.8 of 10 211 WiLM and 76.7% of TMA physicians working 40 hours per week” to “The TMA and WLiM groups had similar work hours, with 74.8(%) of 10 211 WiLM and 76.7% of TMA physicians working 40 hours per week”

Other comments / questions:

Abstract:

Consider stating whether the results found were statistically significant or not within the abstract

Results:

Paragraph beginning “Baseline demographics of female…”

“Early career physicians were more prevalent in non-attendees versus attendee cohorts (less than 5 years in practice 29% vs. 16.5%).”

This statement stood out and if word count allows, there could be additional reflection or exposition on this. Early career physicians are a particularly vulnerable group, with unique stressors, and would benefit from this kind of support, though may be less likely to identify as “ready for leadership” and therefore less likely to self-select.

Paragraph beginning “At baseline…”

“A longitudinal comparison of the 143 women physicians who completed all questions on surveys in 2018 and 2020 demonstrated that participants in WLiM events had lower MBI-EE scores (2.9 at baseline to 2.5 at follow-up) while non-participants showed a less significant decrease from 3.2 at baseline to 3.0 at follow-up. Improvement in single-item MBI-EE scores among the WLiM group was double the improvement seen in female non-participant controls (cohort 1).”

This shows that even without the WLiM program, a woman physician had a 6% decrease in EE compared to a 13.8% decrease in EE for participants - so the proportion “left over” that can be accounted for by the WMiL program is smaller - can the authors comment on whether it is still a significant change?

Paragraph beginning “Participants in WLiM demonstrated an increase in leadership aspirations…”

“Female non-participant controls (cohort 1) experienced the most improvement”

0.36 / 4.8% change for WLiM participants, 0.91 / 17% for cohort 1, 0.11 / 1.7% TMA group

Less improvement in cohort participants for leadership aspiration - wonder why? Interesting to discuss and explore potential factors for this disparity

Discussion:

Paragraph beginning “At baseline, more than ⅓ of all physicians surveyed…”

“The female physicians participating in WLiM events were the only group to demonstrate a statistically significant decrease in emotional exhaustion and, at follow up, WLiM participants demonstrated 13 the lowest MBI-EE scores of all groups surveyed.“

This is notable. I read it twice. “Emotional exhaustion” is such a pervasive and relatable term for women physicians at all stages of career. Again, causes me to reflect on how a program like this might benefit early career physicians (who were less likely to participate in this leadership program)

Paragraph “WLiM participants showed higher leadership aspirations on baseline surveys compared to the other groups. BSWH non-participant female physicians (cohort 1) scored lowest on this domain, which may indicate a bias toward WLiM participation for female physicians who are interested in leadership.”

At first read, this statement seemed to contradict the prior conclusions that cohort 1 participants had the greatest improvement in leadership aspirations. After re-reading, it is clear that the authors are now comparing the baseline numbers and showing how this could be a limitation. Perhaps this would fit better under the paragraph describing “limitations” or be included in the results section to help explain WHY cohort 1 shows more improvement?

Paragraph beginning “A unique strength of this paper…”

“This study evaluated a practical real-world program that can help physician retention…”

This may be misleading. When reviewing the actual data, it appears that retention with organization actually increased MORE for the non-WLiM participants. And if looking at the median value (instead of the mean value), the retention rate of the WLiM group actually got WORSE.

Data table 2

Related to the previous point - the text below the table starts “All parameters (Burnout, leadership aspirations and intention to stay) improved for the women who participated in WLiM events compared to those who did not…” however “intention to stay” did not improve for women who participated in WLiM events COMPARED TO those who did not.

Other potential limitations to address:

The intervention WLiM group had APPs in addition to physicians. By including APPs as participants, this may have an unknown effect or impact on the culture and nature of the programs and the physicians who participated in them

Only binary gender is included in the study and may have limited participation and understanding of non-binary, transgender, or gender-fluid individuals

TMA physicians were older, more experienced, and less likely to have young children. All 3 of these things may have an unknown impact on the outcome of the results

6. PLOS authors have the option to publish the peer review history of their article (what does this mean? ). If published, this will include your full peer review and any attached files.

**Do you want your identity to be public for this peer review?** For information about this choice, including consent withdrawal, please see our Privacy Policy .

Reviewer #1: **Yes: ** Izabella Uchmanowicz

Reviewer #2: No

---

## [Author Response · Author response to Decision Letter 1]

6 Feb 2025

PLOS One Reviewers

RE: PONE-D- 24-35973: Leadership Development as a Novel Strategy to Mitigate Burnout Among Female Physicians

February 4, 2025

Thank you so much for taking the time to review our manuscript and send us your thoughts to enhance the quality of this work and help it be relevant to your audiences. We greatly appreciate your support for promoting evidence-based solutions for the crisis of physician burnout, which disproportionately affects women physicians.

Specific concerns addressed:

Reviewer 1:

1. Despite its strengths, the study has some limitations that could impact its generalizability. The self-selection of participants into the WLiM program introduces a potential bias, as those already motivated to seek leadership roles or address burnout may be overrepresented. Additionally, the study is confined to a specific healthcare system, which may limit its applicability to other institutional or cultural contexts. The reliance on self-reported data further raises concerns about response bias, and the non-randomized design restricts causal inferences. While the manuscript acknowledges these limitations, a more detailed discussion of their implications would enhance the study's transparency and credibility.

Thank you for these recommendations. We have further expanded our discussion of potential biases and limitations with could impact the generalizability of our experience. (second to last paragraph in discussion section, pages 14- 15, lines 323-337). We agree that this should lend to more transparency of our findings and conclusions.

2. External factors, such as concurrent social movements like #MeToo and #TimesUpHealthcare, might have also influenced the outcomes, and the manuscript could benefit from further exploration of these potential confounders. Expanding the analysis to include demographic subgroups could provide additional insights into the intervention's differential impacts. Furthermore, a follow-up assessment of long-term outcomes would be valuable to determine the sustainability of the observed benefits.

We agree that long-term follow up assessments would be of great value as well as further exploration of the external factors of #TimesUpHealthcare and #MeToo for future studies. This study began in 2017, when these provocative programs were in their infancy, and information was not collected about these external factors. We hope to examine the role of these social movements in future studies. For demographic subgroups, we found that non-participation was higher in early career women physicians and that participants who identified as mixed race had some of the highest burnout scores. Detailed subgroup analysis is not possible due to the size of the cohort. We agree that long-term outcomes would be very valuable to determine the sustainability of the observed benefits. We hope to be able to pursue this in a future study.

Have you bought the license for Maslach Burnout Inventory?

Yes, we utilized Funds from The Physicians Foundation to secure the license for the full Maslach Burnout Inventory. We clarified this point in the Survey Measurement’s section of our manuscript (page 7, line 159)

Reviewer #2

1. Minor typos:

Abstract / Results paragraph:

Change “improved levels personal accomplishment…” to “improved levels (of) personal accomplishment…”

Thank you for noting this, modified.

2. Results section / Paragraph beginning “Baseline demographics…”

Change “The TMA and WLiM groups had similar work hours, with 74.8 of 10 211 WiLM and 76.7% of TMA physicians working 40 hours per week” to “The TMA and WLiM groups had similar work hours, with 74.8(%) of 10 211 WiLM and 76.7% of TMA physicians working 40 hours per week”

Thank you for the suggestion, manuscript reflects this change.

3. Paragraph beginning “Baseline demographics of female…”

“Early career physicians were more prevalent in non-attendees versus attendee cohorts (less than 5 years in practice 29% vs. 16.5%).”

This statement stood out and if word count allows, there could be additional reflection or exposition on this. Early career physicians are a particularly vulnerable group, with unique stressors, and would benefit from this kind of support, though may be less likely to identify as “ready for leadership” and therefore less likely to self-select.

We agree that early career physicians are the highest risk for burnout and could benefit greatly from these programs, yet participation is less robust. We have found that being the sole bread winner as well as having young children were potential boundaries to participation. We have added this to the paragraph “WLiM participants showed...” (page 13, lines 291-294). Thank you for highlighting this important group.

4. “A longitudinal comparison of the 143 women physicians who completed all questions on surveys in 2018 and 2020 demonstrated that participants in WLiM events had lower MBI-EE scores (2.9 at baseline to 2.5 at follow-up) while non-participants showed a less significant decrease from 3.2 at baseline to 3.0 at follow-up. Improvement in single-item MBI-EE scores among the WLiM group was double the improvement seen in female non-participant controls (cohort 1).”

This shows that even without the WLiM program, a woman physician had a 6% decrease in EE compared to a 13.8% decrease in EE for participants - so the proportion “left over” that can be accounted for by the WMiL program is smaller - can the authors comment on whether it is still a significant change?

Regarding the decrease in MBI-EE within different groups. Thank you for highlighting such an important part of our data collection. Yes, we can confirm that the decreased of EE specific for the participants of WLiM (relative to non-participants) is independently significant.

5. Paragraph beginning “Participants in WLiM demonstrated an increase in leadership aspirations…”

“Female non-participant controls (cohort 1) experienced the most improvement”

0.36 / 4.8% change for WLiM participants, 0.91 / 17% for cohort 1, 0.11 / 1.7% TMA group

Less improvement in cohort participants for leadership aspiration - wonder why? Interesting to discuss and explore potential factors for this disparity

We are happy to further expand on the magnitude of change on leadership aspirations between groups. The methodology involved a Likert scale. Therefore, when baseline for one group (participants) was anchors towards an extreme (above 7) there is less “room” left for further improvement. Therefore, the continued climb was a positive finding. We believe that events outside the study, such as #MeToo, and #TimesUpHealthcare, also helped to increase leadership aspirations even among non-participants. We hypothesis that the climb was higher within the organization due to more senior physician obtaining visible leadership positions during this period. Thank you for allowing us to explore this.

6. Paragraph beginning “At baseline, more than ⅓ of all physicians surveyed…”

“The female physicians participating in WLiM events were the only group to demonstrate a statistically significant decrease in emotional exhaustion and, at follow up, WLiM participants demonstrated 13 the lowest MBI-EE scores of all groups surveyed.“

This is notable. I read it twice. “Emotional exhaustion” is such a pervasive and relatable term for women physicians at all stages of career. Again, causes me to reflect on how a program like this might benefit early career physicians (who were less likely to participate in this leadership program)

Thank you for taking the time to let this most impactful reality set in. We agree, other studies reveal that EE is the single factor that is most predictive for suicide, maladaptive behavior, divorce and quitting the profession. Thank you for your support as we highlight programs which move this needle and hopefully save lives and careers.

7. Paragraph “WLiM participants showed higher leadership aspirations on baseline surveys compared to the other groups. BSWH non-participant female physicians (cohort 1) scored lowest on this domain, which may indicate a bias toward WLiM participation for female physicians who are interested in leadership.”

At first read, this statement seemed to contradict the prior conclusions that cohort 1 participants had the greatest improvement in leadership aspirations. After re-reading, it is clear that the authors are now comparing the baseline numbers and showing how this could be a limitation. Perhaps this would fit better under the paragraph describing “limitations” or be included in the results section to help explain WHY cohort 1 shows more improvement?

Thank you for this thoughtful suggestion. We did move this discussion to the limitations section as we reflect on the population what was ultimately served with this program design.

8. Paragraph beginning “A unique strength of this paper…”

“This study evaluated a practical real-world program that can help physician retention…”

This may be misleading. When reviewing the actual data, it appears that retention with organization actually increased MORE for the non-WLiM participants. And if looking at the median value (instead of the mean value), the retention rate of the WLiM group actually got WORSE

Thank you for pointing this out. We modified our conclusion to only state the burnout and leadership interests.

9. Data table 2

Related to the previous point - the text below the table starts “All parameters (Burnout, leadership aspirations and intention to stay) improved for the women who participated in WLiM events compared to those who did not…” however “intention to stay” did not improve for women who participated in WLiM events COMPARED TO those who did not.

Thank you for the opportunity to clarify the outcomes. We have modified this text on table 2.

10. Other potential limitations to address:

The intervention WLiM group had APPs in addition to physicians. By including APPs as participants, this may have an unknown effect or impact on the culture and nature of the programs and the physicians who participated in them

Only binary gender is included in the study and may have limited participation and understanding of non-binary, transgender, or gender-fluid individuals

TMA physicians were older, more experienced, and less likely to have young children. All 3 of these things may have an unknown impact on the outcome of the results

Thank you for these thoughtful considerations. We agree that many factors are at play in these populations, due to word count restraints and scope we could not address multiple nuances in this manuscript. The APP data is not included in this analysis. As we started this project in 2017, we did not collect information for non-binary, transgender, and gender-fluid participants, but plan to for future projects. More work is needed to look at physician populations and how to best include and serve everyone to thrive in their careers and lives. We look forward to continuing to explore this in future projects.

Thank you to the editors and reviewers for your very thoughtful and detailed input. We feel that the revisions have strengthened this work and we hope that these changes are acceptable to you and your reviewers. We look forward to hearing from you in the near future and we greatly value your partnership.

Dawn Sears, MD, FACG

---

## [Editor Report · Decision Letter 1]

11 Feb 2025

Leadership Development as a Novel Strategy to Mitigate Burnout Among Female Physicians

PONE-D-24-35973R1

Dear Dr. Sears,

We’re pleased to inform you that your manuscript has been judged scientifically suitable for publication and will be formally accepted for publication once it meets all outstanding technical requirements.

Kind regards,

Francesco Marcatto, Ph.D.

Academic Editor

PLOS ONE

---

## [Editor Report · Acceptance letter]

PONE-D-24-35973R1

PLOS ONE

Dear Dr. Sears,

I'm pleased to inform you that your manuscript has been deemed suitable for publication in PLOS ONE. Congratulations! Your manuscript is now being handed over to our production team.

Kind regards,

on behalf of

Dr. Francesco Marcatto

Academic Editor

PLOS ONE